# The Nucleocapsid Protein of SARS-CoV-2, Combined with ODN-39M, Is a Potential Component for an Intranasal Bivalent Vaccine with Broader Functionality

**DOI:** 10.3390/v16030418

**Published:** 2024-03-08

**Authors:** Yadira Lobaina, Rong Chen, Edith Suzarte, Panchao Ai, Vivian Huerta, Alexis Musacchio, Ricardo Silva, Changyuan Tan, Alejandro Martín, Laura Lazo, Gerardo Guillén-Nieto, Ke Yang, Yasser Perera, Lisset Hermida

**Affiliations:** 1Research Department, China-Cuba Biotechnology Joint Innovation Center (CCBJIC), Lengshuitan District, Yongzhou 425000, China; ylobainamato@ccbjic.com (Y.L.); doris@ccbjic.com (R.C.); aipanchao@ccbjic.com (P.A.); vivian.huerta@cigb.edu.cu (V.H.); alexis.musacchio@cigb.edu.cu (A.M.); rsilva@oc.biocubafarma.cu (R.S.); trim@ccbjic.com (C.T.); 2R&D Department, Yongzhou Zhong Gu Biotechnology Co., Ltd., Yangjiaqiao Street, Lengshuitan District, Yongzhou 425000, China; 3Yongzhou Development and Construction Investment Co., Ltd. (YDCI), Changfeng Industry Park, Yongzhou Economic and Technological Development Zone, No. 1 Liebao Road, Lengshuitan District, Yongzhou 425000, China; 4CIGB: Research Department, Center for Genetic Engineering and Biotechnology, Havana 10600, Cuba; edith.suzarte@cigb.edu.cu (E.S.); alejandro.martin@cigb.edu.cu (A.M.); laura.lazo@cigb.edu.cu (L.L.); gerardo.guillen@cigb.edu.cu (G.G.-N.); 5BCF: R&D Section, Representative Office BCF in China, Jingtai Tower, No. 24 Jianguomen Wai Street, Chaoyang District, Beijing 100022, China

**Keywords:** SARS-CoV-2, broad vaccine, intranasal, ODN, cell-mediated immunity, neutralizing Abs, bivalent vaccine

## Abstract

Despite the rapid development of vaccines against COVID-19, they have important limitations, such as safety issues, the scope of their efficacy, and the induction of mucosal immunity. The present study proposes a potential component for a new generation of vaccines. The recombinant nucleocapsid (N) protein from the SARS-CoV-2 Delta variant was combined with the ODN-39M, a synthetic 39 mer unmethylated cytosine-phosphate-guanine oligodeoxynucleotide (CpG ODN), used as an adjuvant. The evaluation of its immunogenicity in Balb/C mice revealed that only administration by intranasal route induced a systemic cross-reactive, cell-mediated immunity (CMI). In turn, this combination was able to induce anti-N IgA in the lungs, which, along with the specific IgG in sera and CMI in the spleen, was cross-reactive against the nucleocapsid protein of SARS-CoV-1. Furthermore, the nasal administration of the N + ODN-39M preparation, combined with RBD Delta protein, enhanced the local and systemic immune response against RBD, with a neutralizing capacity. Results make the N + ODN-39M preparation a suitable component for a future intranasal vaccine with broader functionality against Sarbecoviruses.

## 1. Introduction

The ongoing SARS-CoV-2 pandemic is a public health crisis that has resulted, to date, in the loss of over 6 million lives and caused an unprecedented disruption to humanity. Laboratories across the globe have worked intensively in developing different vaccines against SARS-CoV-2. In general, most of the approved vaccines have been able to induce more than 50% protection against initial variants, the minimal limit requested by WHO [1,2,3]. Nevertheless, important limitations are identified. First, the safety profile of mRNA vaccines is not similar to those obtained for subunit or inactivated vaccines. Several works are arising with concerns related to this important safety issue [4,5], particularly now with novel nasal-delivered vaccines and the proximity of the blood–brain barrier with its potential vulnerabilities. Second, most of the available vaccines are based on the spike (S) or RBD (receptor binding domain) proteins from the first circulating clades of SARS-CoV-2 [3,5]; therefore, the immunity induced is mainly directed to the close SARS-CoV-2 variants of the Ancestral one [6]. One illustrative example is the limited efficacy of the original mRNA vaccines during large outbreaks of SARS-CoV-2 variants during the first six months of the vaccine rollouts in Northern Hemisphere countries in 2021 [6]. Another example is the limited efficacy of the original mRNA vaccines during large outbreaks of the Omicron/Delta variants [7]. Third, mucosal immunity is not properly induced; consequently, the viral transmission cannot be halted [6,8]. SARS-CoV-2 transmission was initially related to person-to-person spread, but cumulative evidence from sudden emergent COVID-19 epidemics points out the airborne viruses contaminating the environment through persons transferring the pathogen by mouth and nose [9,10]. There is much evidence for airborne carriage for long-distance transportation during prior influenza pandemics [11]. Regardless of the predominant way of viral spread, it is clear that the portal of the entrance is through mucosal routes and that the current vaccines, parenterally administered, do not effectively protect against infection or prevent ongoing transmission [6,7,8].

The mucosal immunity is crucial for SARS-CoV-2 protection. It is recognized that the risk of dying from the COVID-19 respiratory crisis is largely increased in the immune-defenseless elderly and co-morbid subjects in the population [12]. Such vulnerable patients have their innate immunity affected in general [12,13,14], including its mechanisms in the upper respiratory tract [13,14]. On the other hand, the induction of memory pan-specific innate immunity upon oral or intranasal vaccinations has been documented with regular vaccines, indicating the importance of this type of immunity in the mucosal sites [15,16,17]. The low mortality rate observed in children and young people, where the virus infection is no more serious than a coronavirus-induced common cold, could be explained by a more active response of the mucosal innate immunity.

SARS-CoV-2 infection is being considered the third zoonosis event related to human lethal coronaviruses after SARS-CoV in 2002 and MERS-CoV in 2012, although the precise chain of animal-to-human transmission remains undefined [10,18]. The closest SARS-CoV-2 progenitor identified so far, prior to the pandemic, is still only 96.2% similar, suggesting impossible odds of a successful jump to humans. For a theoretical progenitor of 99% similarity the odds are >10^180^ of a zoonotic jump to humans [10]. Further research is therefore required to establish the natural origins of SARS-CoV-2 and related recent coronaviruses [19]. In fact, alternative hypotheses about the origin of the virus are under analysis [9,18,19].

However, several years of research surveying wild animals revealed that bats are infected with the ancestral viruses of SARS, including SARS-CoV-2, and thus, bats serve as a main hypothesized reservoir of many SARS-like viruses [20]. Even more alarming, The USAID PREDICT 1 program (2009–2019) identified 113 novel coronaviruses in animals and people in ecological hotspots with intensive spillover interfaces. Furthermore, the infection of animals with human SARS-CoV-2 may pose a serious problem as recombination events may occur among animal and human coronaviruses to generate novel hybrid viruses with pandemic potential if they spillover in humans with no or partial immunity [20].

The previous facts constitute a real threat to upcoming coronavirus events. The appearance of new variants of SARS-CoV-2 has guided the scientific community to propose a new generation of vaccines with a broader scope of protection: the pancorona vaccines [21,22]. So far, two main approaches are being addressed: multiple immunogenic antigens/regions based on S protein (multivalent) and proteins/designs based on conserved regions among coronaviruses. A multivalent approach requires the presence of various immunodominant regions, and the scope is generally limited to the variants of such regions [23,24,25]. Alternatively, the approach based on conserved antigens could reach a broad scope depending on the level of conservation of the antigen and its proper presentation. The nucleocapsid (N) protein is one example of a conserved antigen that has been tested in different vaccine platforms and combinations with encouraging results against SARS-CoV-2 [26,27,28,29,30,31].

On the other hand, the selection of a suitable adjuvant is crucial in the design of new vaccine formulations. CpG ODNs are considered potent enhancers of the immune response since they bind to and activate Toll-like receptor-9 (TLR9) for initiating an important innate immune response. In fact, more than 100 clinical trials using CpG ODNs have been conducted to assess their use in preventing or treating allergies, infectious diseases, and cancer [32]. In the present work, we combined the recombinant nucleocapsid protein, a conserved antigen from SARS-CoV-2, with the ODN-39M, a synthetic CpG ODN, to favor the formation of aggregate conformations of the N protein based on nucleocapsids natural capacity to bind nucleic acids. It has been previously reported that the ODN-39M is able to interact with other viral capsid proteins and exerts adjuvant effects [33,34,35]. The resulting preparation was evaluated in mice by different administration routes. Interestingly, the N + ODN-39M preparation, administered by the intranasal route, induced the highest anti-N CMI response in the spleen and also elicited humoral immunity in both sera and lungs. Importantly, the overall immunity generated was cross-reactive against the N protein from SARS-CoV-2 Omicron variant, as well as from SARS-CoV-1. Furthermore, a nasal bivalent formulation, based on N + ODN-39M preparation combined with RBD Delta protein, enhanced the local and systemic immune response against RBD with neutralizing capacity and modulation toward a Th1-like pattern. These findings support the use of the N + ODN-39M preparation as a potential component of a future intranasal vaccine with broader functionality.

## 2. Materials and Methods

### 2.1. Recombinant Proteins, Peptide and ODN-39M

The recombinant antigens were purchased from Sino Biological Inc. (Beijing, China), including N proteins from SARS-CoV-2, Delta variant (40588-V07E29), Omicron variant (40588-V07E34), SARS-CoV-1 (40143-V08B), MERS-CoV (40068-V08B), HCoV-229E (40640-V07E), and RBD proteins from SARS-CoV-2: Delta variant (40592-V08H90), Ancestral variant (40592-VNAH), and ACE-2-His (10108-H08B).

The peptide N_351–365_ from SARS-CoV-2 (ILLNKHIDAYKTFPP) was synthesized with ≥97% purity by Zhejiang Peptides Biotech (Hangzhou, China).

The ODN-39M, a 39 mer, whole phosphodiester backbone ODN (5′-ATC GAC TCT CGA GCG TTC TCG GGG GAC GAT CGT CGG GGG-3′), was synthesized by Sangon Biotech (Shanghai, China).

### 2.2. In Vitro Aggregation Procedure of Nucleocapsid Protein with ODN-39M

The N protein from SARS-CoV-2, Delta variant, was subjected to in vitro aggregation, as previously described [33,34], with few modifications. Briefly, in a 100 µL reaction, 40 µg of N protein was mixed with 60 µg of ODN-39M in Tris/EDTA buffer, pH 6.9. The mixtures were incubated for 30 min at 30 °C in a water bath and, after, were stored at 4 °C for 4 h. Finally, each preparation was centrifuged at 14,000× *g* for 10 min. The resulting supernatant was collected and tested for protein concentration.

### 2.3. Immunization Experiments

Adult (6 to 8 weeks old) female Balb/c mice (inbred, H-2d) were housed at Beijing Vital River Laboratory Animal Technology Co., Ltd. (Beijing, China). The standard of laboratory animal room complied with the national standard of the people’s Republic of China GB14925-2010. All the experimental protocols were approved by the Institutional Animal Care and Use Committee.

Animals were randomly distributed in groups of five to six animals each and immunized with three doses, administered on days 0, 7, and 21 by intranasal (i.n) or subcutaneous (s.c) routes, according to each experimental design. Formulations for s.c administration were prepared with aluminum hydroxide (alum; Alhydrogel, Invitrogen, Waltham, MA, USA) as adjuvant at a final concentration of 1.4 mg/mL. A dose of 10 μg of each protein per mouse was evaluated. All the immunogens were dissolved in sterile PBS. For i.n and s.c administrations, the immunogen was administered in a final volume of 50 μL and 100 μL, respectively. In each experiment, placebo-immunized groups were included as controls by each route.

In the first experiment, six groups, each containing six mice, were employed. Groups 1 and 2 received N protein and N + ODN-39M, respectively, adjuvated with alum by the s.c route. Groups 3 and 4 were intranasally immunized with N and N + ODN-39M, respectively. Groups 5 and 6 acted as controls receiving PBS + alum (s.c) and PBS (i.n), respectively. Three animals per group were sacrificed on days 12 and 18 after the last immunization.

For the second mouse experiment (n = 5/group), the animals from Groups 1 and 2 were intranasally immunized with N and N + ODN-39M, respectively. Group 3 received PBS (i.n). Animals were sacrificed on day 26 after the last dose.

In the third experiment (n = 5/group), Groups 1 and 2 received N and N + ODN-39M, respectively. Groups 4 and 3 received the same formulations as 1 and 2, respectively, but included RBD. Groups 5 and 6 were immunized with RBD and PBS, respectively. All groups were intranasally immunized. Animals were sacrificed 12 days after the last immunization.

A fourth experiment was conducted using five animals per group and exploring a different administration schedule. All groups were intranasally immunized on days 0, 15, and 30. Groups 1 and 2 received monovalent formulations based on RBD and N + ODN-39M, respectively. Group 3 was immunized with the bivalent formulation N + ODN-39M + RBD, and Group 4 received PBS. Animals were sacrificed 30 days after the last dose.

At the indicated time points, three types of samples were collected: sera, bronchoalveolar fluid (BALF), and spleens.

### 2.4. Assessment of Humoral Immune Response by ELISA and a Surrogate Virus Neutralization Test

The antibody response in sera and BALF was monitored by ELISA. Briefly, anti-IgG, subclasses and -IgA ELISAs were carried out as previously described [36]. Briefly, 96 well high-binding plates (Costar, Washington, DC, USA) were coated with N (3 μg/mL) or RBD (2 μg/mL) proteins and blocked with 2% skim milk solution. Samples were evaluated in duplicates using different dilutions starting from 1/50 for sera. BALF were assayed directly without dilution. Specific horseradish peroxidase conjugates (Sigma, St. Louis, MO, USA) were employed, and OPD (Sigma, USA)/hydrogen peroxide substrate solution was used. After 10 min of incubation in the dark, the reaction was stopped using 2 N sulfuric acid and the optical density (O.D) was read at 492 nm in a multi-plate reader (FilterMax F3, Molecular Devices, San Jose, CA, USA). For antibody response measured in sera data was represented in the graphics as log10 titers. The arbitrary units of titers were calculated by plotting the O.D values obtained for each sample in a standard curve (a hyper-immune serum of known titer). The positivity cut-off was established as two times the average of O.D obtained for a pre-immune sera pool. On the other hand, for antibody response in BALFs, since the positive signals obtained are usually lower, results were represented as O.D at 492 nm.

For detecting functional antibodies generated by RBD formulations, a surrogate virus neutralization test (sVNT) was used since it allows determining the capacity of mice sera to inhibit the interaction of RBD with ACE2 protein. Mouse sera were assayed using the SARS-CoV-2 Inhibitor Screening ELISA kit (Sino Biological, KIT001), according to the supplier instructions, to evaluate the inhibitory activity against the Ancestral strain. In this assay, antibodies present in sera compete with ACE2 for binding to RBD (Ancestral variant), coating the wells of the plate. Bound ACE2 is detected via the polyhistidine tag using an anti-polyhistidine Mab conjugated to HRP. In this experimental set up, the signal decreases as increments the inhibitory capacity of the serum.

For detecting inhibitory antibodies against the RBD Delta variant, an in-house sVNT was developed following the same design but using RBD from the Delta variant as a coating protein. Samples, positive and negative controls were diluted 1:25, followed by two serial dilutions 1/3 or 1/5 with PBS 0.3% BSA, 0.05% Tween 20. The dilutions were combined with an equal volume of ACE2-His in a dilution plate. Fifty microliters of each mixture of ACE2 and serum dilution were added to separate RBD-coated wells of a 96-well plate (ThermoScientific 442404, Waltham, MA, USA) and incubated at 25 °C for 1 h. Bound ACE2 was detected with the conjugate anti-His tag HRP (SB 105327-MM02TH) and developed using OPD/H_2_O_2_ as substrate. Absorbance was read at 492 nm on an ELISA microplate reader.

In both assays, binding inhibition was calculated as (1 −  OD value of sample/OD value of Negative control)  × 100%. Inhibitory titer was defined as the dilution in which each animal serum inhibits 20% of the binding of ACE2 without any competitor, calculated by linear regression using GraphPad Prism.

### 2.5. Assessment of Cellular Immune Response by IFN-γ ELISPOT

IFN-γ ELISPOT assay was performed using a Mouse IFN-γ ELISpot antibody pair (Mabtech, Stockholm, Sweden). Splenocytes were isolated in an RPMI culture medium (Gibco, New York, NY, USA). Samples (three or five mice per group) were processed individually, with the exception of the control groups (Placebos), which were processed as pooled samples of three randomly selected mice. Duplicate cultures (5 × 10^5^ and 1 × 10^5^ splenocytes per well) were settled at 37 °C for 48 h, at 5% CO_2_, in a 96-well round-bottom plate with 10 µg/mL of N_351–365_ peptide or N protein, 10 µg/mL of concanavalin A (ConA), or medium. After, the whole content of this plate was transferred to an ELISPOT pre-coated plate and incubated at 37 °C for 16–20 h, at 5% CO_2_. The incubation conditions for conjugated antibodies and the following steps were taken as recommended by the manufacturers. A stereoscopic microscope (AmScope SM-1TSZ, Irvine, CA, USA) coupled to a digital camera was used for spot count.

### 2.6. Pseudotyped VSV-Based Neutralization Assay

The assay to quantify the neutralizing capacity of the sera was carried out as previously described [37]. Briefly, a commercial kit containing a viral stock of vesicular stomatitis virus (VSV) pseudotyped with the S protein from SARS-CoV-2 Ancestral strain and a Luciferase substrate solution (Darui Biotech, Guangzhou, China) was used. In a 96-well culture plate (Costar, USA), different sera dilutions were incubated with the recommended concentration of virus for 1 h at 37 °C and 5% CO_2_. A column of the plate was reserved for virus control (VC, without sera sample), and another column for cell control (CC, without virus). After, 2 × 10^4^ Huh-7 cells (provided by Darui Biotech, China) were added per well, and the plate was incubated for 24 h at 37 °C and 5% CO_2_. Once finished, the incubation, 150 uL of supernatant from each well was removed, and 100 uL of Luciferase substrate solution was added per well. After 5 min of incubation in the dark, the content of each well was resuspended and transferred to a white 96-well plate (Costar, USA). The luminescence was read using a FilterMax F3 microplate reader (Molecular Device, USA). The calculation of the inhibition percentage was done using the recommended formula:

Inhibition rate = (1 − average of luminescence for sample—average luminescence for CC)/(average luminescence for VC—average luminescence for CC) × 100%

In the assay, positive and negative controls were included. The positive control was a commercial neutralizing antibody against Spike protein from SARS-CoV-2 Ancestral strain (Sino Biological, Beijing, China) and a pool of mice sera with a known neutralizing titer 1:4000. In turn, the negative control was the pool from placebo groups. All the quality criteria recommended for this kind of assay were properly accomplished.

For determining the neutralizing antibodies titers (EC_50_) vs. SARS-CoV-1 in sera from the bivalent formulation immunized group, a professional service was hired by the CRO (Darui Biotech, China). The pseudotyped VSV system, carrying the Spike protein from SARS-CoV-1, was used for the determinations. A positive control established at the CRO for this assay (sera with known neutralizing titer) was included. A pool from the mice placebo group was used as a negative control. All the quality criteria recommended for this kind of assay were properly accomplished. The neutralizing antibody titer (EC_50_) was calculated using the Reed-Muench method. 

### 2.7. Statistical Analysis

For statistical analyses, the GraphPad Prism version 5.00 statistical software (Graph-Pad Software, San Diego, CA, USA) was used. Antibody titers were transformed to log10 for a normal distribution. For the non-sero-converting sera, an arbitrary titer of 1:50 was assigned for statistical processing. One-way Anova test, followed by Tukey’s post-test, were used as parametric tests for multiple group comparisons. In the case of non- parametric multiple comparisons, the Kruskal–Wallis test and Dunn’s post-test were employed. A standard *p*-value consideration was as follows: ns = no significance, *p* > 0.05; *, *p* < 0.05; **, *p* < 0.01; ***, *p* < 0.001.

## 3. Results

### 3.1. The N + ODN-39M Combination, Administered by Intranasal Route, Is Immunogenic in Balb/C Mice

A first mouse experiment was conducted to explore the immunogenicity of the N + ODN-39M combination administered by intranasal and subcutaneous routes (Figure 1a). Three doses of each immunogen were administered based on the positive results obtained in a previous exploratory mouse experiment for the N + ODN-39M combination intranasally inoculated (Appendix A). Particularly, for subcutaneous immunization, alum was added to the formulation based on the positive results of the ODN-39M and alum combination for dengue antigens [33,34]. As shown in Figure 1b, all groups receiving the N protein by subcutaneous route induced high levels of anti-N antibodies in sera. In contrast, by the intranasal route, only the group that received N + ODN-39M was able to induce a positive response, with titers similar to those obtained by the subcutaneous route. The IgG1 and IgG2a subclasses against N Delta protein were also determined (Figure 2c,d). The IgG1 levels induced by G4 (N + ODN-39M, intranasally administered) were significantly lower (*p* < 0.01) compared to those elicited by Groups 1 and 2 (subcutaneously inoculated). On the contrary, the IgG2a levels were significantly higher in the G4 (*p* < 0.001). Together, these results strongly suggest that a Th1 pattern was induced in mice receiving the formulation N + ODN-39M by intranasal route, as confirmed by the IgG1/IgG2a ratio = 1 for this group (Appendix A).

The IgA antibodies against N Delta were measured in BALF. As expected, only G4 (N + ODN-39M) elicited a positive response (Figure 1e). In turn, the anti-N IgG pattern of response found in BALF replicates the one observed in serum (Appendix A). Probably due to the lack of adjuvant in the formulation, the N protein alone did not induce mucosal immunity.

To evaluate the CMI, the frequency of IFN-γ secreting spleen cells was evaluated after in vitro stimulation with two agents: the conserved peptide N_351–365_ and the N Delta protein. For the first stimulation agent, two different determinations were completed on days 12 and 18, respectively (Figure 2a,b). At both times, a similar pattern of CMI response was observed for groups G1, G2 and G4. A detectable IFN-γ secreting cell response was obtained in three out of three animals only from G4, whereas no positive response was seen for animals from G3 (N protein in PBS, intranasal administered). In addition, only one out of three mice elicited a positive response in each group immunized by subcutaneous route (G1 and G2). On the other hand, when the whole N protein was used as a stimulating agent, the overall behavior was similar; however, the number of responder animals in the groups subcutaneously immunized was increased to three out of three. Consistent with the previous determinations, the G4 (N + ODN-39M, intranasally administered) showed a clear trend to generate the highest response (Figure 2c).

### 3.2. Intranasally Administered N + ODN-39M Preparation Induces a Cross-Reactive Immune Response against N Protein until the Sarbecovirus Level

Aiming to assess the cross-reactive scope of the immunity generated by the combination N + ODN-39M administered by the intranasal route, a second mouse experiment was conducted (Figure 3a). In this experiment, the time point selected was 26 days after the last dose to explore the duration of the immunity at this point. Again, the N + ODN-39M preparation and not the N protein alone, both administered by the intranasal route, induced anti-N mucosal and systemic immunity. As shown in Figure 3b, the group inoculated with the combination N + ODN-39M induced in sera a positive IgG response against the N protein from SARS-CoV-2 Delta variant, SARS-CoV-2 Omicron variant and SARS-CoV-1. On the other hand, none of the groups showed a detectable response against N proteins from MERS-CoV and HCoV-229E (Appendix A). A similar pattern of response was obtained when the IgA was measured in BALF samples against the different N proteins (Figure 3c). 

To test the cross-reactive CMI, the frequency of IFN-γ secreting spleen cells was measured upon in vitro stimulation with (1) the conserved peptide N_351–365_, (2) N protein from SARS-CoV-1, (3) N protein from MERS-CoV, and (4) N protein from HCoV-229E. In line with the humoral immune response, animals receiving the intranasal administration of N + ODN-39M preparation exhibited a positive response against the peptide N_351–365_ and the N protein from SARS-CoV-1 (Figure 3d), whereas no response was detected for MERS-CoV and HCoV-229E proteins (Appendix A). Results confirmed the cross-reactive nature of the CMI induced by N + ODN-39M preparation until Sarbecovirus subgenus level. Again, the N + ODN-39M preparation and not the N protein alone, both administered by the intranasal route, induced anti-N mucosal and systemic immunity.

### 3.3. The N + ODN-39M Preparation Exerts an Adjuvant Effect on RBD Protein When Both Components Are Intranasally Co-Administered

A third mouse experiment was conducted to explore the immunogenicity of a nasal bivalent formulation comprising N + ODN-39M + RBD (Figure 4a). The RBD was selected as a potential inductor of neutralizing Abs. For anti-N response in sera, as shown in Figure 4b, G2 and G3 induced significant levels of Abs compared to the control group G6 (*p* < 0.001). The groups that received N protein without ODN-39M (G1 and G4) did not induce a positive response. The anti-N IgG levels elicited by G3 (bivalent formulation) were high (˃10^3^) yet statistically lower than those raised by G2 (*p* < 0.001).

Regarding the IgG antibody response in sera against RBD, only the group intranasally immunized with the bivalent formulation N + ODN-39M + RBD (G3) showed a 100% seroconversion and IgG titers statistically higher than the rest (*p* < 0.001) (Figure 4c). On the other hand, the group that received N + RBD (G4) showed a 60% seroconversion (with titers ≤10^3^), and in the group immunized with RBD in PBS, no animal seroconverted. These results provide the first evidence of the adjuvant effect of the N + ODN-39M combination over RBD by intranasal route.

In line with previous results, the groups G2 and G3 generated in sera statistically similar anti-N IgG1 and IgG2a levels, suggesting a Th1 pattern (Figure 4d,f).

In turn, the pattern of IgG subclasses anti-RBD (Figure 4e,g) revealed higher IgG1 titers for G3 (bivalent formulation). In addition, only G3 was able to generate an IgG2a positive response when compared to the rest of the groups (*p* < 0.001), suggesting again a modulation toward a Th1 pattern.

The assessment of IgA antibodies in BALF against both antigens is shown in Figure 5a,b. In both cases, only the groups inoculated with formulations containing the N + ODN-39M preparation elicited a positive response of IgA antibodies. Similar to the behavior observed for the anti-N IgG response generated in sera, the IgA response in BALF tended to be higher in the group receiving the monovalent formulation (G2) compared with the bivalent formulation (G3).

The results of CMI testing are shown in Figure 5c. Upon stimulation with the conserved N_351–365_ peptide, IFN-γ secreting cell response was statistically similar between G2 and G3, whereas no response was seen among splenocytes from the rest of the groups. In parallel, when splenocytes were stimulated with RBD Delta protein, a positive response was only obtained in two out of five animals from the group that received the bivalent formulation N + ODN-39M + RBD (Figure 5d).

To explore the scope of the humoral immune response elicited by the bivalent formulation, sera and BALFs were also evaluated against the heterologous antigens: N from SARS-CoV-1 and RBD from Ancestral and Omicron variants of SARS-CoV-2. Sera exhibited Ab titers against N from SARS-CoV-1 (Figure 6a and Appendix A) and RBD from the Ancestral variant of SARS-CoV-2 (Figure 6b), following a pattern similar to that obtained for homologous antigens. A positive response was also obtained against RBD from the Omicron variant (Appendix A). On the other hand, in BALFs, three out of five animals induced IgA Abs against RBD from the Ancestral variant of SARS-CoV-2, whereas all animals were positive against the homologous antigen (Figure 6c).

In order to test the broad functionality of Abs elicited in sera and BALFs against RBD in the group receiving the bivalent formulation, a surrogate virus neutralization assay (sVNT) was employed using the RBD from Delta and Ancestral variants of SARS-CoV-2. This ELISA assay measures the capacity of the Abs to inhibit the interaction between RBD and the cell receptor ACE2.

As shown in Figure 6d,e, all animals elicited antibodies with inhibitory activity against both variants. As expected, the higher values were obtained for the homologous RBD and in sera samples. Finally, a neutralization assay based on the VSV pseudotyped virus system, carrying the Spike protein of the heterologous strain (Ancestral variant), was also employed. As a result, in the group immunized with the bivalent formulation, five out of five sera were positive with more than 50% viral inhibition at a dilution of 1:50, two out of five at 1:150 dilution, and only one out of five at 1:450 (Figure 6f).

### 3.4. The Bivalent Formulation, N + ODN-39M + RBD, Induces Neutralizing Abs in Sera and Mucosal Samples, and Anti-RBD CMI

To induce higher levels of neutralizing Abs and increase the number of responders against RBD by CMI, an additional study was conducted evaluating the bivalent formulation following a more extended immunization schedule. In this protocol, the three doses were administered on days 0, 15, and 30.

Concerning the anti-N humoral immunity, in sera and BALFs, high levels of anti-N IgG and IgA, respectively, were obtained in the bivalent formulation group (Figure 7a,b). Importantly, no statistical differences were detected between the monovalent (N + ODN-39M) and bivalent formulations (N + ODN-39M + RBD) in BALFs. Similarly, no differences in the percentage of responders were observed between these two groups by ELISPOT upon in vitro stimulation with the N protein (Figure 7c).

The anti-RBD immunity is shown in the down panel of Figure 7. As expected, high levels of anti-RBD IgG and IgA Abs were obtained in sera and BALFs, respectively, in the group receiving the bivalent formulation (Figure 7d,e). Again, the adjuvant effect of the N + ODN-39M component over the RBD humoral immunity was demonstrated with significant differences between the monovalent (RBD) and the bivalent (N + ODN-39M + RBD) groups. Interestingly, the RBD-specific CMI response generated in the present study by the group receiving the bivalent formulation was higher compared with our previous data since 100% of the evaluated mice responded positively (Figure 7f). On the other hand, corresponding with previous results, no responders were detected for the monovalent group (RBD), confirming the adjuvant effect of N + ODN-39M over RBD immunity in terms of CMI.

Finally, the neutralizing activity of individual sera samples and pooled BALFs from the group immunized with the bivalent formulation was tested by the VSV pseudotyped virus system, carrying the Spike protein of the heterologous strain (Ancestral variant). As shown in Figure 8a,c, four out of five sera have neutralizing titers higher than 1:500. Three of them exhibited titers higher than 1:1250. The BALF pool also shows a positive neutralizing activity (>50% of inhibition at 1:20 dilution) versus the SARS-CoV-2 heterologous strain (Figure 8b). On the other hand, the anti-SARS-CoV-1 neutralizing capacity of the sera generated after immunization with the bivalent formulation was also evaluated using the VSV system pseudotyped with S protein from SARS-CoV-1. Three out of five evaluated sera show a positive neutralizing response with EC_50_ titers higher than 50 (Figure 8c); these sera match with those showing the higher neutralizing response versus SARS-CoV-2 Ancestral strain.

## 4. Discussion

In the last two years, a number of studies have shown that N protein is a target of cellular immunity in humans, and such a response has been correlated with protection against severity [38]. The study published by Matchett et al., 2021, demonstrated that the N protein, presented in the Ad5 platform, induced humoral and cellular immunity in mice, and such response correlated with protection against SARS-CoV-2 challenges [39]. Dangi et al., 2021, also showed that combining spike and nucleocapsid vaccine formulations improved distal protection in the brain [40]. These authors even demonstrated the protective role of anti-N Abs, proven by passive transfer experiments [41].

However, despite the encouraging results validating the N protein as an appealing component for a vaccine against SARS-CoV-2 [42], two questions remain unanswered: (1) the breadth of the immunity generated and (2) its capacity to be immunogenic as a recombinant protein, by intranasal administration using adjuvants suitable for human use. Both attributes are highly desirable in a novel coronavirus vaccine to halt, since the beginning, the spreading of current variants of concern or even new zoonotic events.

The present work provides evidence to answer the two aforementioned questions. The N Delta variant, as a recombinant construct obtained in *Escherichia coli*, was selected as an antigen since the Delta variant was the major circulating strain when these studies started. Despite the Delta variant being overtaken by subsequent variants, this variant was considered one of the most aggressive strain variants of SARS-CoV-2, and its immunogenicity has not been deeply explored in the context of vaccine candidates. On the other hand, subunit-based vaccines have important advantages over the rest of the vaccine platforms in terms of safety and easier storage conditions.

In turn, the ODN-39M is a whole phosphodiester ODN containing CpG motifs with a proven adjuvant capacity for other viral vaccine candidates [33,34,35,43]. Given that the fundamental function of the SARS-CoV-2 N protein is to package the viral genome into a ribonucleic nucleoparticle, we hypothesized that the N protein could interact with the ODN-39M to form aggregate structures, as referred for viral genomic RNA both in vitro and in mammalian cells [44].

For the first immunological evaluation in mice, we explored the N + ODN-39M preparation by two administration routes: intranasal and subcutaneous. Among the intranasal groups, only the group receiving N + ODN-39M elicited high levels of anti-N IgG Abs in sera, similar to those elicited by the groups inoculated with the N in alum by subcutaneous route. In addition, only this group elicited anti-N IgA Abs in BALFs, and the pattern of IgG subclasses revealed a typical Th1 response. This behavior is consistent with the results obtained in the CMI assay, where only the group immunized with N + ODN-39M by the intranasal route induced a positive response in all animals tested when the peptide N_351–365_ was used as a stimulating agent. These experimental pieces of evidence highlight the key role of the N + ODN-39M preparation in the induction of a mucosal humoral anti-N response, as well as the CMI against the peptide N_351–365_. Of note, this peptide spans a conserved region among sarbecovirus which is immunodominant in SARS-CoV-2 Balb/C infected mice. In addition, the Venezuelan equine encephalitis replicon particles vector (VRP), expressing this single SARS-CoV-2–specific CD4+ T cell epitope, partially protected mice from SARS-CoV-2 infection four weeks after VRP/peptide vaccination, as determined by moderate reduced viral titers and diminished lesions in the lungs. This experiment provided direct evidence for supporting the protective role of memory T-cells against the conserved peptide N_351–365_ [45].

Two elements could contribute to the results obtained with the nasal N + ODN-39M preparation: (1) the adjuvant role of the ODN-39M and (2) the route of administration. It is well known that CpG ODNs are considered very promising adjuvants [46]. The ODN-39M evaluated in the present work is also a CpG ODN but has a phosphodiester backbone, making the formulation N + ODN-39M very attractive for human use since the thioate backbone has been associated with adverse reactions in therapeutic interventions [46]. We consider that the presence of ODN-39M in the formulation potentiates the immune response against N protein, as reported for other intranasal antigens [47]. On the other hand, the intranasal administration favors the antigen presentation in mucosal tissues, which are enriched with plasmacytoid dendritic cells that are easily activated by TLR9 agonists as the CpGs ODNs [48]. Of note, despite the N protein’s ability to interact with ODNs and the known immunogenicity of aggregated antigens [49] in the present work there is no evidence that ODN-39M aggregation to the N protein is critical to the immunogenicity of the vaccine preparation.

Based on the obtained results, the nasal N + ODN-39M preparation was selected for the second mouse experiment to evaluate the breadth of the immunity induced. In general, the cross-reactivity obtained is in line with the levels of homology of SARS-CoV-2 N protein among the coronavirus tested. Within the sarbecovirus genus, it is estimated a range of 87–99% of identity whereas, for N MERS-CoV and N HCoV-229E proteins, their identity with the nucleocapsid of SARS-CoV-2 is 48% and lower than 38%, respectively [50]. Although the response obtained did not reach the level of the Betacoronavirus genus, where MERS-CoV is representative, we consider that the cross-immunity reached is highly valuable. Several samplings for coronaviruses have been conducted in East and South East Asia, and around 50 SARS-related coronaviruses have been detected across 10 species of bat [51]. In fact, authors found that bat-borne SARS-related coronaviruses present a particular pandemic threat due to the extraordinary viral genetic diversity represented among several sympatric species of their horseshoe bat hosts. Accordingly, Crook JM et al., 2021, asserted that the highest the probability for homologous recombination of sarbecoviruses through co-infection, the biggest the possibility of novel zoonotic emergence. Thus, co-infection of horseshoe bats with their natural suites of coronaviruses and with SARS-CoV-2 could lead to the development of a novel zoonotic emergence [52].

Two additional mouse experiments were addressed to assess the nasal bivalent formulation, composed of the N + ODN-39M preparation and the RBD from the Delta variant as an inductor of neutralizing Abs. The recombinant RBD fragment is a component of at least five approved vaccines for emerging use, such as the Cuban vaccines, Abdala and Soberana Series (01, 02 and Soberana Plus) [53,54], and ZF001, a Chinese vaccine approved in China and Uzbekistan [55]. Such vaccines have been capable of controlling the magnitude of the different infection’s waves of SARS-CoV-2 variants; nevertheless, as for the other approved vaccines, their capacity to halt the virus transmission has been limited [56,57].

In the present work, results demonstrated the adjuvant effect of the combination N + ODN-39M over the mucosal/systemic humoral response and cell-mediated immune response, all against RBD. Such adjuvant effect, under the Th1 pattern, was more evident in the mouse experiment where immunogens were administered 15 days apart. Interestingly, the anti-RBD immunity generated by the bivalent formulation did not affect the mucosal Abs and CMI against the N protein. However, in sera, there was a decrease in anti-N Abs, which indicates that some level of antigenic competence took place between the two proteins. Despite this, we consider that the resultant balance in sera is positive since anti-RBD Abs are more relevant as they can initially neutralize the virus. Of note, the intranasal administration of the bivalent formulation induced neutralizing Abs in sera and BALF, measured by sVNT and VSV pseudotyped virus carrying the Spike protein of Ancestral variant, both systems widely used [58,59]. Surprisingly, an additional neutralizing response was detected against SARS-CoV-1, particularly when the bivalent formulation was administered two weeks apart, providing important evidence that N + ODN-39M, administered by intranasal route in the context of the bivalent formulation with RBD, potentiates the induction of neutralizing Abs against not only SARS-CoV-2 but also cross-reactive ones at sarbecovirus level.

Considering that our bivalent vaccine candidate is administered by the intranasal route, the levels of homologous neutralizing Abs induced are relevant despite the fact that they may not be comparable to titers reported, for instance, for parenteral RBD-based vaccines that employed higher doses or dimers of RBD, and alum as a Th2-prompt adjuvant [60,61]. For the particular case of the BALF samples, the effect of dilution should be taken into account since 1 mL of PBS is used for washing mice’s airway mucosa.

In general, the magnitude of the immune response elicited by the bivalent formulation could be increased in the future by three alternative ways: (1) Exploring higher RBD doses in the formulation, (2) Using the intranasal vaccine as a booster in previously vaccinated and/or infected population, consequently increasing the magnitude and spectrum of the neutralizing response in both compartments, and (3) Exploring the combination of the nasal route with a parenteral one, using the same bivalent formulation. As results obtained for other vaccines, the route combination would result in additive or synergistic effects that could potentiate the neutralizing Abs response [36,62,63].

Together, we consider that the combination of N + ODN-39M constitutes a promising nasal vaccine component that can be added to the list of enhancers of RBD immune response by this route [64,65,66,67,68]. Of note, despite the positive immunogenicity data obtained in the present work, based on potential markers of protection even with a broad scope, we acknowledge that its functionality in preventing infection and transmission against sarbecoviruses needs to be further elucidated in the relevant animal model under BSL-3 conditions. Once the protective capacity is defined, the safety of the bivalent formulation should be carefully assessed, particularly for testing the impossibility of crossing the blood–brain barrier and, consequently, not provoking neurological damage. Of note, some intranasally administered vaccines have been previously registered, showing a safe profile [69]. The most successful example is FluMist, a live attenuated Influenza vaccine, in the market for more than 10 years and extensively administered in adults and childrens [70]. On the other hand, intranasal vaccines based on protein subunits represent a lower risk of potential neurological damage; this is the case of HeberNasvac, a novel therapeutic vaccine for chronic hepatitis B infection, registered some years ago with a good safety profile in different populations [71].

## 5. Conclusions

N + ODN-39M preparation, administered by intranasal route, is able to induce an anti-N cross-reactive immunity at systemic and mucosal compartments, reaching Sarbecovirus level. In addition, when this preparation is mixed with RBD, forming the bivalent vaccine preparation, it potentiates the immune response to RBD antigen with neutralizing capacity, supporting its use as a potential component of a future pancorona vaccine with broader functionality. Particularly, the nasal bivalent formulation (N + ODN-39M + RBD) constitutes a very promising vaccine candidate as a booster dose to amplify and broaden previous SARS-CoV-2 immunity generated by either natural infection or vaccine, especially one generated by inactivated vaccines.

## Figures and Tables

**Figure 1 viruses-16-00418-f001:**
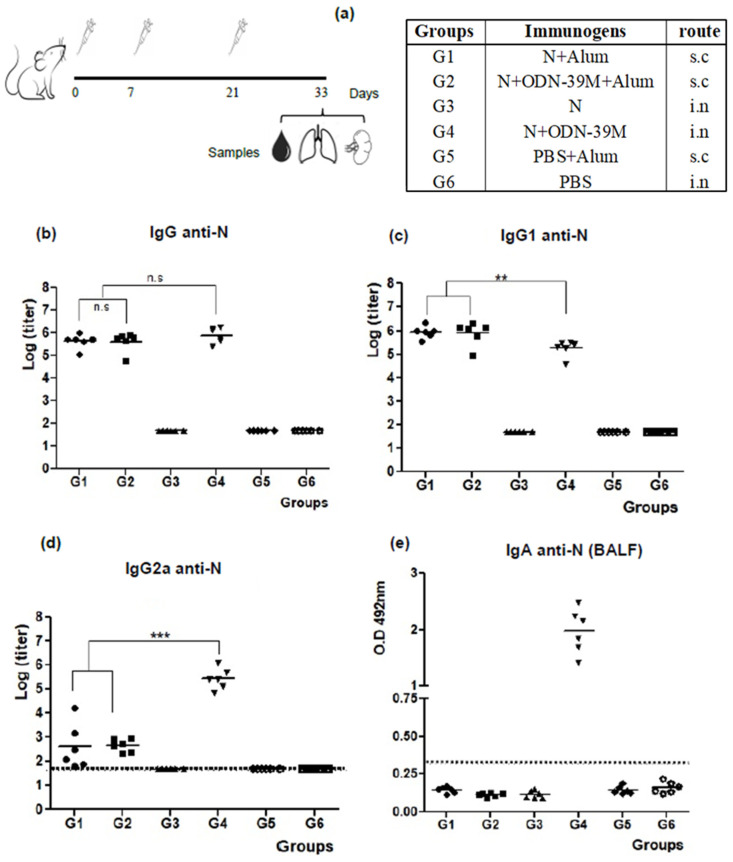
Evaluation of humoral immunity elicited by SARS-CoV-2 nucleocapsid (N) based formulations. (**a**) Diagram of Balb/c mouse immunization. Six- to eight-week-old mice were immunized with three doses of each formulation according to the following group design: G1: N + Alum, sc. G2: N + ODN-39M + Alum, sc. G3: N (PBS), in. G4: N + ODN-39M, in. G5: PBS + Alum, sc. G6: PBS, in. sc: subcutaneous, in: intranasal. Twelve days after the third immunization, mice were sacrificed, and antibody responses in the serum and bronchoalveolar lavage fluid (BALF) were evaluated (n = 6 animals per group). (**b**–**d**) The systemic humoral immune response was measured in sera by (**b**) IgG ELISA against N, (**c**) IgG1 ELISA against N and (**d**) IgG2a ELISA against N. Data are represented as the log of the titers. The statistical analysis was done by One-Way Anova, followed by Tukey’s multiple comparison test. ** *p* < 0.01, *** *p* < 0.001. (**e**) The mucosal humoral immune response measured in BALFs by IgA ELISA against N. Data are expressed as O.D, and the horizontal bar represents the mean. The dotted line indicates the limit of positive response.

**Figure 2 viruses-16-00418-f002:**
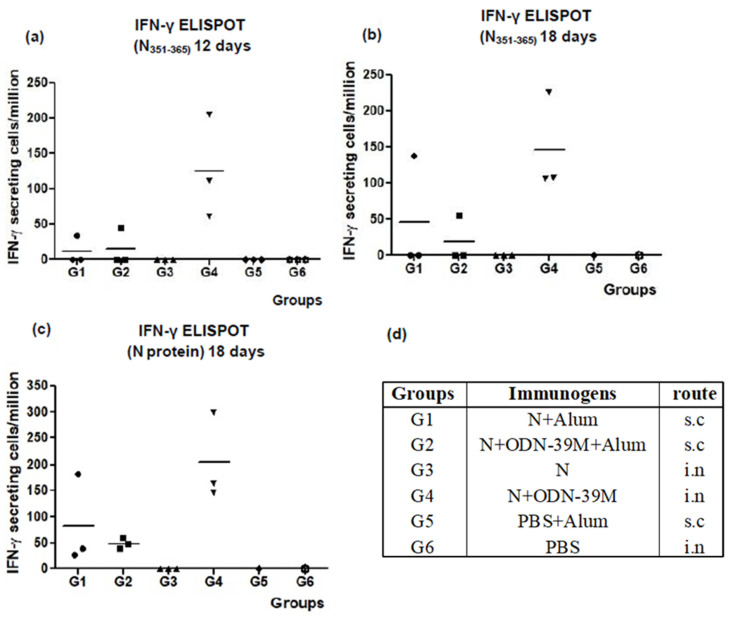
Cell-mediated immune responses induced in Balb/c mice. Six- to eight-week-old mice were immunized with three doses of each formulation, at days 0, 7, and 21, according to the following group design: G1: N + Alum, sc. G2: N + ODN-39M + Alum, sc. G3: N (PBS), in. G4: N + ODN-39M, in. G5: PBS + Alum, sc. G6: PBS, in. sc: subcutaneous, in: intranasal. Twelve days (**a**) or 18 days (**b**,**c**) after the third immunization, mice were sacrificed, and spleens were extracted and in vitro stimulated with both N 351–365 peptide (**a**,**b**) and N protein (**c**), and the frequency of the resulting IFN-γ secreting cells was measured by ELISPOT. The graphs show the results for n = 3 individual mice, and the horizontal bar represents the mean. (**d**) Description of the groups.

**Figure 3 viruses-16-00418-f003:**
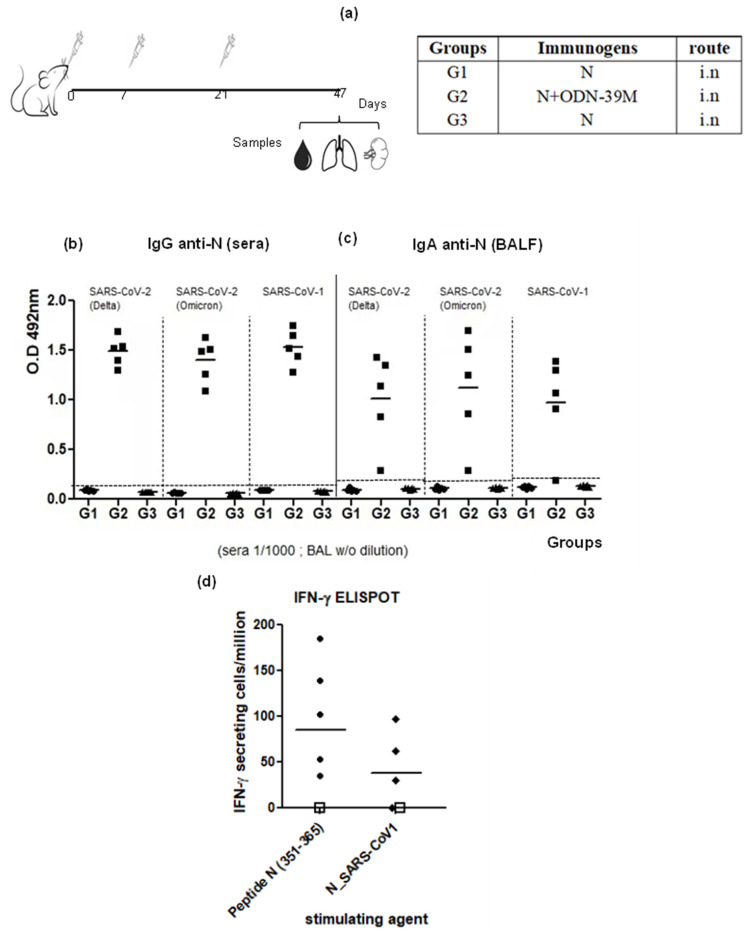
Cross-reactivity of the immune response generated by the intranasal administration of N + ODN-39M. (**a**) Diagram of Balb/c mouse immunization. Six- to eight-week-old mice were immunized with three doses of each nasal formulation according to the following group design: G1: N protein, G2: N + ODN-39M and G3: PBS. Twenty-six days after the third immunization, mice were sacrificed, and antibody responses against N protein from the SARS-CoV-2 Delta variant, SARS-CoV-2 Omicron variant and SARS-CoV-1 were measured in (n = 5 mice per group), by (**b**) IgG ELISA in sera, 1:1000 dilution and (**c**) IgA ELISA in bronchoalveolar lavage fluid (BALF), without dilution. Data are expressed as the O.D values. The dotted lines indicate the limit of positive detection and the horizontal bar represents the mean. (**d**) Cell-mediated immune response. Twenty-six days after the third immunization, mice were sacrificed, and spleens were extracted and stimulated in vitro with both N_351–365_ peptide and N protein from SARS-CoV-1. The frequency of the resulting IFN-γ secreting cells was measured by ELISPOT. The graph represents n = 5, or 4, individual mice, respectively. The empty square symbol represents a pool of three placebo mice.

**Figure 4 viruses-16-00418-f004:**
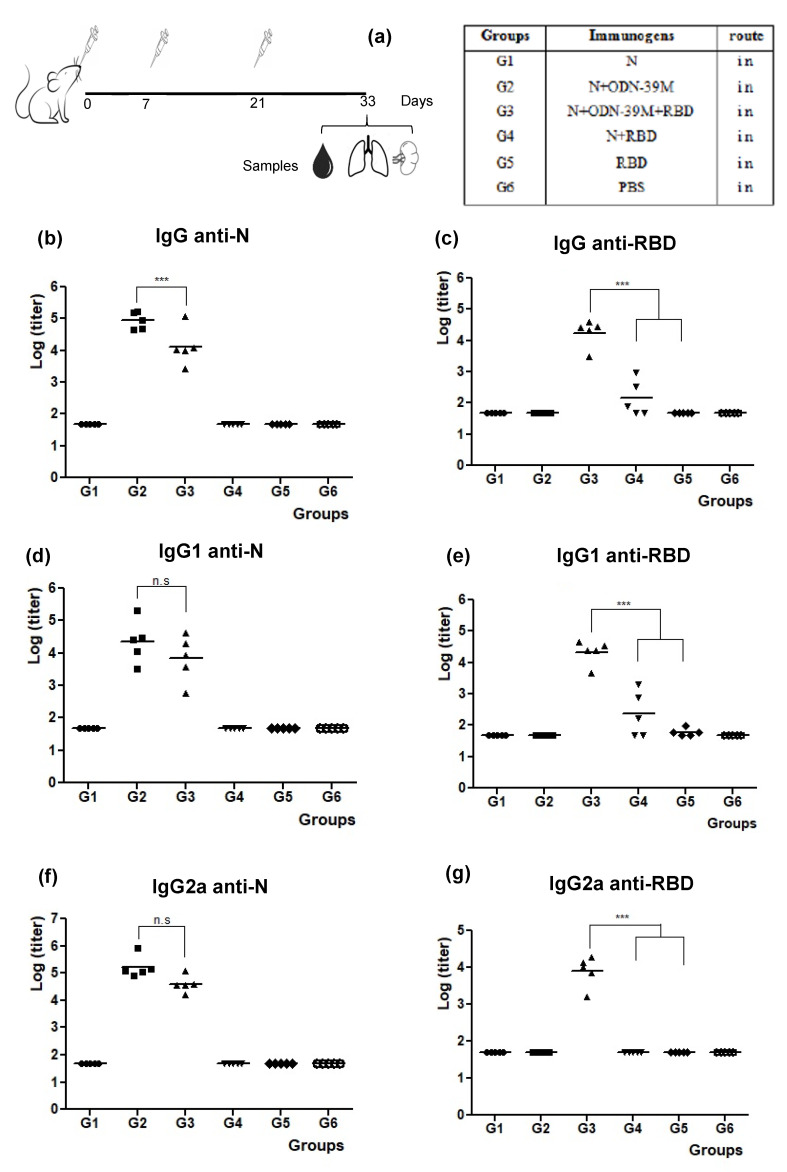
The pattern of the humoral immunity against N and RBD proteins in sera. (**a**) Diagram of Balb/c mouse immunization. Six- to eight-week-old mice were immunized with three doses of each nasal formulation according to the following group design: G1: N protein, G2: N + ODN-39M, G3: N + ODN-39M + RBD, G4: N + RBD, G5: RBD, G6: PBS. Twelve days after the third immunization, mice were sacrificed, and antibody responses in the serum were evaluated in n = 5 mice per group by (**b**) IgG ELISA against N, (**c**) IgG ELISA against RBD, (**d**) IgG1 ELISA against N, (**e**) IgG1 ELISA against RBD, (**f**) IgG2a ELISA against N, (**g**) IgG2a ELISA against RBD. Data are expressed as the log of the titers, and the horizontal bar represents the mean. The one-way Anova test followed by Tukey’s post-test were used as parametric tests for multiple group comparisons *** *p* <0.001.

**Figure 5 viruses-16-00418-f005:**
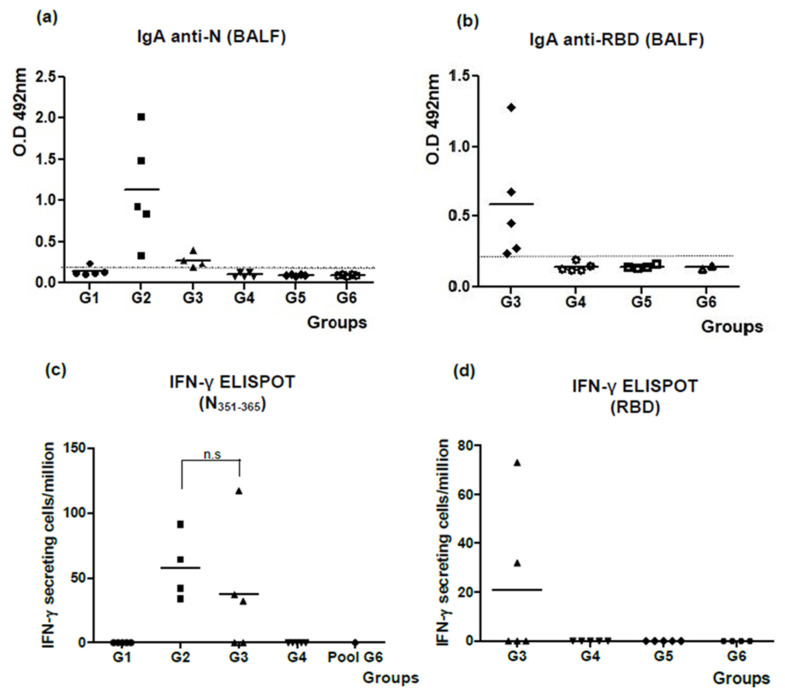
Evaluation of mucosal humoral immunity and cell-mediated immunity for the bivalent nasal formulation. Six- to eight-week-old Balb/c mice were immunized with three doses of each nasal formulation at 0, 7, and 21 days, according to the following group design: G1: N protein, G2: N + ODN-39M, G3: N + ODN-39M + RBD, G4: N + RBD, G5: RBD, G6: PBS. Twelve days after the third immunization, mice were sacrificed, and antibody responses in BALFs (without dilution) were measured against (**a**) N and (**b**) RBD. Data are expressed as O.D values, and the horizontal bar represents the mean. The dotted line indicates the limit of positive detection. The same day after the third immunization, for measuring cell-mediated immune response, spleens were extracted and stimulated in vitro with both N_351–365_ peptide (**c**) and RBD (**d**). The frequency of the resulting IFN-γ secreting cells was measured by ELISPOT. Graphs represent values (n = 4 or 5). For statistical analysis, the Kruskal–Wallis test and Dunn’s post-test were employed.

**Figure 6 viruses-16-00418-f006:**
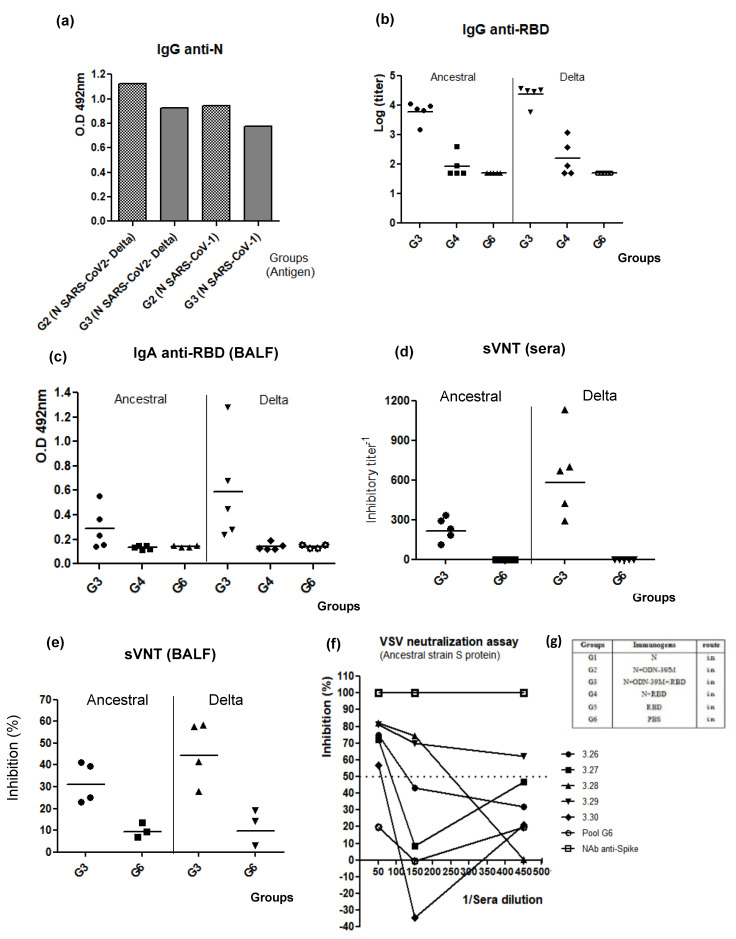
Evaluation of the cross-reactivity of humoral immunity induced by the bivalent formulation N + ODN-39M + RBD. Six- to eight-week-old Balb/c mice were immunized with three doses of each nasal formulation at 0, 7, and 21 days. Group design: G1: N protein, G2: N + ODN-39M, G3: N + ODN-39M + RBD, G4: N + RBD, G5: RBD, G6: PBS. Sera from groups G2, G3, G4 and G6 were evaluated by (**a**) IgG ELISA against N proteins from SARS-CoV-2 Delta variant and SARS-CoV-1 (pooled samples), (**b**) IgG ELISA against RBD Ancestral and Delta variants. (**c**) BALFs from groups G3, G4 and G6 were evaluated by IgA ELISA against RBD Ancestral (Wuhan strain: YP_009724390.1) and Delta variants. Graphics for ELISAs represent either the O.D values or the log of titers. In panels b, c, and d, five animals per group are represented. sVNT against RBD Ancestral and Delta variants in sera (**d**) and BALF (**e**) (dilution 1:2). Inhibitory titer was defined as the highest dilution in which each serum inhibits more than 20%. (**f**) Neutralization test by pseudotyped VSV system, carrying the Spike protein from Ancestral variant. The serum of each animal from G3 was evaluated at different dilutions. As controls in the assay, a pool of serum from the placebo group (G6) was included as a negative control, and a commercial neutralizing Ab (NAb) anti-spike protein from SARS-CoV-2 Ancestral strain as a positive control. Dotted lines correspond to 50% of inhibition. (**g**) Description of the groups.

**Figure 7 viruses-16-00418-f007:**
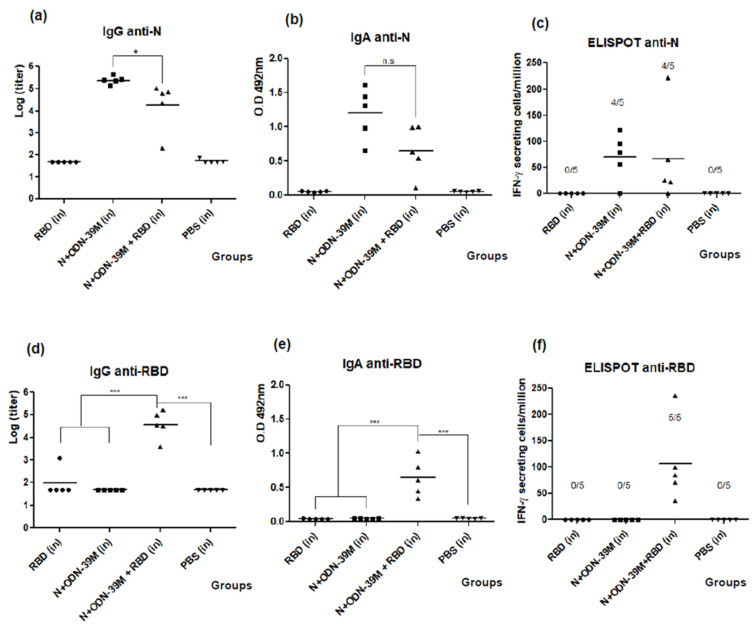
Evaluation of humoral immunity and cell-mediated immunity for the bivalent nasal formulation. Six- to eight-week-old Balb/c mice were immunized with three doses of each nasal formulation at 0, 15, and 30 days. After the third immunization, mice were sacrificed, and antibody responses in BALFs and sera, respectively, were measured against N (up panel) and RBD (down panel). (**a**,**d**) Data are expressed as the log of the IgG titers. (**b**,**e**) Data are expressed as O.D values. The horizontal bar represents the mean. The One-way Anova test and Tukey’s post-test were used as parametric tests for multiple group comparisons. ns, *p* > 0.05; *, *p* < 0.05; ***, *p* < 0.001. The frequency of IFN-γ secreting cells was measured by ELISPOT. Spleen cells were isolated from five mice per group and in vitro stimulated with N (**c**) and RBD (**f**) proteins. In the graphs, the horizontal bar represents the mean, and the frequency of responders is shown for each group.

**Figure 8 viruses-16-00418-f008:**
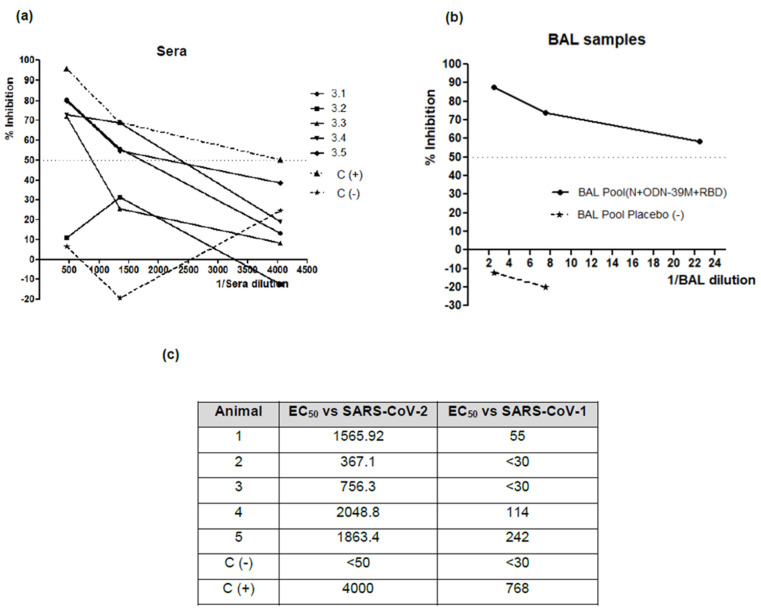
Neutralizing antibodies induced by the (N + ODN-39M + RBD) bivalent formulation. A neutralization test was performed by a pseudotyped VSV system carrying the Spike protein from the Ancestral variant of SARS-CoV-2 (**a**,**b**). The individual serum and a pooled BALF sample from the group receiving the bivalent formulation were evaluated. A pool of serum or BALF from the placebo group was included as a negative control (C(-)), and a pool of sera with a known anti-SARS-CoV-2 neutralizing titer was used as a positive control (C(+)). (**c**) Table representing the EC_50_ titers for the bivalent formulation group sera samples, tested by the pseudotyped VSV system, carrying the Spike protein from SARS-CoV-2 Ancestral strain and SARS-CoV-1.

## Data Availability

Data are contained within the article and Appendix A.

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
