# Peer review of "The Nucleocapsid Protein of SARS-CoV-2, Combined with ODN-39M, Is a Potential Component for an Intranasal Bivalent Vaccine with Broader Functionality"

_viruses, 2024, doi:10.3390/v16030418_

Round 1
Reviewer 1 Report
Comments and Suggestions for Authors
All the important issues that need to be addressed are in the uploaded file, viz vaccine safety issues, vaccine efficacy, zoonotic origins, human engineered etc, mode of spread P-to-P versus non P-to-P . The ethical issue noted above re vaccine safety intra-nasal route needs to be addressed.

Comments on the Quality of English LanguageThe minor issue only of use of word 'mice' in the context of a 'mouse group' i.e use of 'mice' group is wrong in this context. But otherwise the English was good and clear
Author Response
Thank you very much for your detailed and valuable revision of the MS. Concerning all the theoretical issues such as the limitations of mRNA vaccines, the origin of the virus, and the way of virus spreading, they were deeply analyzed and additional paragraphs were added. Please, find attached our response to your revision.

Reviewer 2 Report
Comments and Suggestions for Authors
The authors present a comprehensive, well-designed and well-written work regarding an intranasal bivalent vaccine against SARS-CoV-2 using the nucleocapsid and RBD protein. The study´s methodology is sound and its design and testing are elaborate, the manuscript is written in a comprehensible way and it is easy to read. However, there are a few minor points that need to be addressed:
1. The antigen constructs were obtained in Escherichia coli if I understand correctly? Please elaborate possible post-translational changes to the molecules, which are different in eukoryotic and prokaryotic cells, such as glycosylation.
2. Please explain the optimal pH for ODN-39M in its relation to stability and immunogenicity.
3. Line 37: an “s” is missing. Please put “different vaccine” in plural.
4. Line 125: “Groups 4 and 3 received the same formulations than 1 and 2”- please correct “than” to “as”.
5. Line 488: Please write Escherichia coli (E. coli) all in Italics.
6. The quality of Figure 1 (a), 2 (d), 3 (a), 4(a), 6 (f) are below average as they obviously are screenshots from a word document, where one can still see the “Spell-Check”.
7. Please cite the following research papers:
Ahlen G, Frelin L, Nikouyan N, Weber F, Hoglund U, Larsson O, Westman M, Tuvesson O, Gidlund EK, Cadossi M, Appelberg S, Mirazimi A, Sallberg M, Consortium O (2020) The SARS-CoV-2 N Protein Is a Good Component in a Vaccine. J Virol
Smits VAJ, Hernández-Carralero E, Paz-Cabrera MC, Cabrera E, Hernández-Reyes Y, Hernández-Fernaud JR, Gillespie DA, Salido E, Hernández-Porto M, Freire R (2021) The Nucleocapsid protein triggers the main humoral immune response in COVID-19 patients. Biochem Biophys Res Commun
Hevesi Z, Gerges D, Kapps S, Freire R, Schmidt S, Pollak D, Schmetterer K, Frey T, Lang R, Winnicki W, Schmidt A, Harkany T, Wagner L (2022) Preclinical Establishment of a Divalent Vaccine against SARS-CoV-2. Vaccines
Author Response
Thank you very much for your revision and valuable comments/suggestions. Please, find attached our answers

Reviewer 3 Report
Comments and Suggestions for Authors
In this study, the efficacy is evaluated of an intranasally administered bivalent vaccine comprised of the nucleocapsid protein of SARS-CoV-2 in combination with CpG ODN, in addition to the RBD Delta protein. CpG ODNs are considered powerful enhances of the immune response and this is certainly conformed here, as evidenced on many levels. First, this formulation induced a high anti-N CMI response, as well as a potent humoral immune response in both sera and lungs. Second, the bivalent formulation combined with RBD Delta induced a local and systemic immune response against RBD in a Th1-like pattern.
This is considered a well-designed and appropriately interpreted study that makes an exceptionally strong case for the potential for the use of N+ODN-39M in combination with the RBD Delta protein for intranasal administration to control SRS-CoV-2. A major strength of the study is its demonstrated cross-reactivity against the N protein of both the Delta and Omicron variants. The data are all very strong and convincing, as well as being subjected to appropriate statistical analysis. There are no detected weaknesses in any aspect of the study.
One very minor criticism is that term CpG ODN should be defined better for the reader when first used in the manuscript.
Comments on the Quality of English LanguageMinor problems with English usage.
Author Response
Thank you very much for your revision and valuable comments. Please, find attached our answer:

Round 2
Reviewer 1 Report
Comments and Suggestions for Authors
Report revised version Lobaina et al 2024
This is a better MS. Corrections to text and correction of reference numbers has been the main task of this reviewer. The aim is to publish a paper with more qualified and balanced assumptions on vaccine safety and SARS-CoV-2 origins and assumed lethality.
This report is using the revised Reference numbers in the revised Manuscript, which are in black ink and red ink for additional references added. So that there is no misunderstanding of what reference numbers I am referring to the Revision list is attached in the uploaded combined PDF file, entitled" "Report revised version Lobaina et al 2024 + Ref list.pdf ".
1. Lines 49-59 did not seem to make sense . This reviewer had to re-write the sentences and assign correct references
The current text is below and does not make sense in English in many parts and references appear to be wrongly assigned:
Despite their undoubtedly relevant role they have two commonNevertheless, important limitations. are identified. First, the safety profile of mRNA vaccines is not similar to those obtained for subunit or inactivated vaccines, in fact, several works are arising with concerns related on this important safety issue [4]. Second, most of the available vaccines are based on the spike (S), or RBD (receptor binding domain), proteins from the first circulating clades of SARS-CoV-2 [45]; therefore,the immunity induced is mainly directed to the close SARS-CoV-2 variants of the Ancestral one [5]. Second6]. One illustrative example is the limited efficacy of the original mRNA vaccines during large outbreaks of the Omicron/Delta variants [7]. Third, the mucosal immunity is not
This is re-written to improve clarity and with correct reference assignment thus:
Despite their undoubtedly relevant role they have common yet nevertheless important limitations that are identified [4-8]. First, the safety profile of mRNA vaccines is not similar to those obtained for subunit or inactivated vaccines [6], in fact, several works are arising with concerns related on this important safety issue of mRNA vaccines [5], particularly now with novel nasal delivered vaccines and the proximity of the blood brain barrier with its potential vulnerabilities. Second, most of the available vaccines are based on the spike (S), or RBD (receptor binding domain), proteins from the first circulating clades of SARS-CoV-2 [4,6]; therefore, the immunity induced is mainly directed to the close SARS-CoV-2 variants of the Ancestral one [4,6]. Second, one illustrative example is the limited efficacy of the original mRNA vaccines during large outbreaks of SARS -CoV-2 variants during the first 6 months of the vaccine rollouts in Northern Hemisphere countries during 2021 [8]. Another illustrative example is the limited efficacy of the original mRNA vaccines during later large outbreaks of the Omicron/Delta variants [7]. Third, the mucosal immunity is not
2. Lines 59-68 are edited and modified, the original is:
Third, the mucosal immunity is not properly induced; consequently, the viral transmission cannot be halted [6]. 8]. SARS-CoV-2 transmission was initially related to person to person spread but cumulative evidences, from suddenly emergent COVID-19 epidemics, points out to the airborne viruses contaminating the environment from which persons transfer the pathogen to mouth and nose [9-11]. Regardless the predominant way of viral spread, it is clear that the portal of entrance is through mucosal routes and for instance, the current vaccines, parenteral administered, do not effectively protect against infection or prevent ongoing transmission [7].
This has been re-written with appropriate reference numbers thus:
Third, the mucosal immunity is not properly induced; consequently, the viral transmission cannot be halted [8,9]. SARS-CoV-2 transmission was initially related to person to person spread but cumulative evidences, from suddenly emergent COVID-19 epidemics, points to the airborne viruses contaminating the environment from which implies persons transfer the pathogen to mouth and nose [10,11]. There is much evidence for the airborne carriage for long distance transportation during prior influenza pandemics [12]. Regardless the predominant way of viral spread, it is clear that the portal of entrance is through mucosal routes and for instance, the current vaccines, parenteral administered, do not effectively protect against infection or prevent ongoing transmission [7-9].
3. Lines 69-70 is edited as 'zoonosis' cannot be proven, it is changed from
SARS-CoV-2 infection is the third zoonosis related to human lethal coronaviruses after SARS-CoV in 2002 and MERS-CoV in 2012.
To
SARS-CoV-2 infection is the third recent pandemic related to human lethal coronaviruses after SARS-CoV in 2002 and MERS-CoV in 2012.
4. Lines 70-72 have been edited as the text is a bit confusing because of incorrect references from
The mucosal immunity is crucial for SARS-CoV-2 protection. It is recognized that the risk of dying from the COVID-19 respiratory crisis is largely increased in the immune defenseless elderly and co-morbid subjects in the population [12]. Such vulnerable patients have their innate immunity affected in general [13, 14], including its mechanisms in upper respiratory tract [12]. On the other hand, it has been documented the induction of memory pan-specific Innate Immunity upon oral or intranasal vaccinations with regular vaccines, indicating the importance of this type of immunity in the mucosal sites [15-17]. The low mortality rate observed in children and young people, where the virus infection is no more serious than a coronavirus-induced common cold, could be explained by a more active response of the mucosal innate immunity.
To a more correct version with corrected references
The mucosal immunity is crucial for SARS-CoV-2 protection. It is recognized that the risk of dying from the COVID-19 respiratory crisis is largely increased in the immune defenceless elderly and co-morbid subjects in the population [13]. Such vulnerable patients have their innate immunity affected in general [13-15], including its mechanisms in upper respiratory tract [13,15]. On the other hand, it has been documented the induction of memory pan-specific Innate Immunity upon oral or intranasal vaccinations with regular vaccines, indicating the importance also of this type of immunity in the mucosal sites [16-18]. The low mortality rate observed in children and young people, where the virus infection is no more serious than a coronavirus-induced common cold, could be explained by a more active response of the mucosal innate immunity.
5. Lines 83-90 have been edited as the text is a bit confusing because of unproven assumptions and incorrect references from. These improvements are shown in " To a more correct version with corrected references" below. The current text Line 83-90 is:
SARS-CoV-2 infection is being considered the third zoonosis event related to human lethal coronaviruses after SARS-CoV in 2002 and MERS-CoV in 2012 although the precise chain of animal-to-human transmission remains undefined [18]. This gap has fostered alternative hypothesis about the origin of the virus [19]. However, several years of research surveying wild animals revealed that bats are infected with the ancestral viruses of SARS including SARS-CoV-2 and, thus, bats serve as a main reservoir of many SARS-like viruses [20].
To a more correct version with corrected references is:
SARS-CoV-2 infection is being considered the third putative zoonosis event related to human lethal coronaviruses after SARS-CoV in 2002 and MERS-CoV in 2012 although the precise chain of animal-to-human transmission remains undefined [11,19]. This gap has fostered alternative hypothesis about the origin of the virus [11,19,20]. However, several years of research surveying wild animals revealed that bats are infected with the ancestral viruses of SARS including SARS-CoV-2 and, thus, bats serve as a main hypothesizeed reservoir of many SARS-like viruses yet the closest SARS -CoV-2 progenitor prior to the pandemic is still only 96.2% similar suggesting impossible odds of a successful jump to humans. For a theoretical progenitor of 99% similarity the odds are > 10180 of a zoonotic jump to humans [11]. Further research is therefore required to establish the natural origins of SARS-CoV-2 and related recent coronaviruses [20].
6. Lines 93-97 have been edited as the text again is a bit confusing and incorrect references from
These facts,Furthermore, infection of animals with human SARS-CoV-2 may pose a serious problem as recombination events may occur among animal and human coronaviruses to generate novel hybrid viruses with pandemic potential if they spillover in humans with no or partial immunity [20].
To a more correct version with corrected references
Furthermore, infection of animals with human SARS-CoV-2 may pose a serious problem as recombination events may occur among animal and human coronaviruses to generate novel hybrid viruses with pandemic potential if they spillover in humans with no or partial immunity [21].
7. Lines 98-112 have been edited as the referencing is incorrect, from
The previous facts constitute a real threat for upcoming coronavirus events, which along with the appearance of new variants of SARS-CoV-2, has guided to the scientific community to propose a new generation of vaccines with broader scope of protection: the pancorona vaccines [7, 821, 22]. So far, two main approaches are being addressed, multiple immunogenic antigens/regions based on S protein (multivalent) and proteins/designs based on conserved regions among coronaviruses. A multivalent approach requires the presence of various immunodominant regions, and the scope is generally limited to the variants of such regions [9–1123–25]. Alternatively, the approach based on conserved antigens could reach a broad scope depending on the level of conservation of the antigen and its proper presentation. The nucleocapsid (N) protein is one example of conserved antigen which has been tested in different vaccine platforms and combinations with encouraging results against SARS-CoV-2 [12-1426-31].
To this with correct references
The previous facts constitute a real threat for upcoming coronavirus events. The appearance of new variants of SARS-CoV-2 has guided to the scientific community to propose a new generation of vaccines with broader scope of protection: the pancorona vaccines [22,23]. So far, two main approaches are being addressed, multiple immunogenic antigens/regions based on S protein (multivalent) and proteins/designs based on conserved regions among coronaviruses. A multivalent approach requires the presence of various immunodominant regions, and the scope is generally limited to the variants of such regions [23–25]. The alternative approach is based on conserved antigens and could reach a broad scope depending on the level of conservation of the antigen and its proper presentation. The nucleocapsid (N) protein is one example of conserved antigen which has been tested in different vaccine platforms and combinations with encouraging results against SARS-CoV-2 [26-31].
8. Line 119 the reference has to change from [1532] to [32]
9. Line 119 the reference has to change from [16-1833-35] to [33-35].
10. Line 153 the reference has to change from [1633, 34] to [33, 34]
11. Line 184 micemouse should be mouse . Please ensure that other corrections from "mice" to "mouse" are also made.
12. From Line 207 I assume the authors will correct the red ink reference number as the correct one. But please check again, given the issues already identified above, that the correct reference number is included.
13. New text Line 787-799, new text on safety is approved but please double check references.
Just minor correction of text here
Once defined the protective capacity, the safety of the bivalent formulation should be carefully assessed, particularly for testing the impossibility of crossing the blood brain barrier and consequently, no provoking neurological damage.
To
Once the protective capacity is defined, the safety of the bivalent formulation should be carefully assessed. In particular for testing the impossibility of crossing the blood brain barrier and consequently, not provoking neurological damage.

Comments on the Quality of English LanguageAuthor Response
Thank you very much for the second revision of the MS. We have added the sentences suggested by the reviewer (in red) and the references were corrected, (also in red). Of note, the word: References was numbered by the reviewer therefore, the original numbering was moved. We kept the original numbering.